# Managing Knowledge Resources in Family Firms: Opportunity or Challenge?

**Omar Belkhodja** 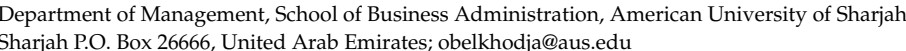

Department of Management, School of Business Administration, American University of Sharjah, Sharjah P.O. Box 26666, United Arab Emirates; obelkhodja@aus.edu

**Abstract:** The purpose of this study was to explore the specificities of the relationship between knowledge management (KM) processes and the potential and realized absorptive capacities in the context of a knowledge-based view. The paper advances our understanding of the contributions of knowledge management processes and the potential and realized absorptive capacities in small- and medium-sized family firms. We draw on case studies of two small- and medium-sized family businesses operating in different industries. Our results show that the choice of the KM approach and the family business characteristics determine the extent to which a family firm is successful in managing its knowledge processes and absorptive capacity. Moreover, the results indicate that family businesses are impacted by their own characteristics, such as the fact that they do not dissociate between the personalities of the owners and the business, and are context-specific. Since the focus of this research was limited to KM processes and absorptive capacity, it would be beneficial for future research to investigate the mechanisms that enable firms to manage their potential and realized absorptive capacities and the extent to which they generate dynamic capabilities through KM processes. Further studies of the impacts of family business characteristics on the firm's success in managing knowledge resources are also recommended.

**Keywords:** knowledge management; knowledge processes; absorptive capacity; family business; case study

## 1. Introduction

Over the past few decades, knowledge has become the major source of competitive advantage for businesses [1,2] as firms compete in a complex and competitive environment, in which customers increasingly seek value [3]. The knowledge-based view introduces a shift in the value creation process as it acknowledges that knowledge structures have inherent value creation capabilities [4] and that intangible resources have replaced tangible ones in the process of value creation [5]. Based on the knowledge-based view, the alignment and integration of knowledge resources with business strategy are necessary for knowledge value creation [6,7]. Knowledge resources need to be deployed and managed through appropriate processes [8] to standardize and formalize knowledge flows and improve production activities [9]. Knowledge processes are defined as the knowledge means by which value is added throughout a company's activities to create a competitive advantage [10].

Knowledge management research has focused extensively on how large firms manage knowledge-based resources and implement effective strategies to leverage newly acquired or already-existent knowledge. However, only a few studies have addressed how small- and medium-sized enterprises, that operate under more resource constraints, take advantage of their knowledge resources and manage them to gain a competitive advantage. Small- and medium-sized family businesses face even more challenges when managing their knowledge resources due to their unique characteristics compared to non-family businesses such as their ownership structure, strategic intent, and the influence of the attitude and behavioral traits of the owner-managers on the strategic direction adopted by the family

business. The fact that family firms represent over 75% of registered businesses in most economies, the unique set of skills and resources available in family businesses, and the lack of studies targeting knowledge management processes and strategies in small- and medium-sized enterprises set the context of this research, where we explore the specificities of family businesses in terms of knowledge management processes and absorptive capacities. We define a family firm as a business with two generations of the same family and where strategies are influenced by family members.

An in-depth analysis of the knowledge management and family business literature allowed us to identify the following three gaps:

1.  Many studies highlight the importance of KM processes and the need for companies to focus on value creation [11,12]. Extant research in the KM field has separated between the study of KM processes and a firm's knowledge absorption capacities. Past research overlooked the impacts of the firm's ACAP on the process of value creation [13].
2.  Most KM efforts are fragmented [12] and research is often limited to the study of externally acquired knowledge or internally created knowledge. Only a few studies analyzed the KM processes from the perspective of both externally and internally generated knowledge in the context of the knowledge-based view.
3.  Despite their major contribution to the economy and an increased scholarly interest, we still know very little about KM processes in small- and medium-sized family firms and how these family firms manage their knowledge resources and their absorptive capacities. There is a lack of KM studies that are applied to family businesses despite their unique configuration of human capital and unique approach in managing knowledge-based resources.

This paper aims to contribute to the literature on knowledge management and family business and addresses the research gaps identified above by examining the KM processes and ACAPs in two small- and medium-sized family businesses. Our contribution consists of advancing the understanding of knowledge mechanisms by focusing on the study of the intricacies that exist between the KM processes and the firm's absorptive capacity. Hence, the objectives of our study are (1) to explore the specificities of the relationship between KM processes and a firm's absorptive capacity in the context of a knowledge-based view using externally and internally generated knowledge as input; this requires clarifying the role of knowledge processes in the company's value creation dynamics and exploring the relationships between the processes and capacities mentioned above; (2) to delineate and characterize the importance of the potential and realized absorptive capacities; and (3) to examine and explore the knowledge specificities of small- and medium-sized family businesses based in the United Arab Emirates (UAE). Two case studies in the context of UAE-based family businesses are used as the main research method.

The remainder of this paper is organized as follows. First, we provide the literature background of our research focus and key concepts investigated. Then, we detail our methodology and the case study approach adopted to collect the data and conduct case study analyses. We wrap up the paper with a discussion of our contributions along with a discussion of future avenues for research.

## 2. Literature Background

### 2.1. Knowledge Management Processes

2.1.1. Knowledge Acquisition

Knowledge acquisition refers to the action of acquiring external knowledge to renew a firm's existing routines and generate dynamic capabilities [14,15]. The action of acquisition requires prior recognition of the value of external knowledge for the firm's operations [16]. Often, firms fail to recognize the value of knowledge-based resources because of their embedded knowledge resources and rigid capabilities [9,16,17]. Value recognition starts from the firm's existing capabilities and routines and the assessment of current customer demand [18]. However, not all knowledge acquisition efforts are fruitful. The quality of

acquisition capabilities depends on the intensity and speed with which the firm identifies the valuable external knowledge and gathers it from these sources [15]. Moreover, acquisition efforts must be directed to activities that sustain the firm's competitive advantage and support the constant renewal of dynamic capabilities. Family businesses dedicate fewer resources to knowledge acquisition as they are less growth-oriented and more conservative in approaching strategic decisions. The centralization of decision-making authority concentrates decisions, including those related to knowledge acquisition, in the hands of a few family members who manage the firm based on personal judgement and views rather than an objective assessment of the company knowledge resource needs. "Familiness", which contributes to create the family business's unique identity, favors internal knowledge transfer among family members rather than efforts targeting the acquisition of new knowledge from external sources [16].

Knowledge transfer, which is the action of making knowledge available to others in an organizational context [19], is an essential mechanism of KM [20,21]. When there is insufficient background information, a lack of shared language, and a lack of common interests between the sender and the recipient of knowledge [22], the transactional approach of knowledge transfer emphasizing the outcome of the transfer process is used [23]. However, as family businesses possess a strong sense of identity, a unique social system, and "familiness", the collaborative approach which promotes shared perceptions and views and an active collaboration in the process of knowledge transfer is used to transfer tacit knowledge. Convergence in cognitive maps in the family business leads to the use of a joint process of knowledge transfer and calls for the establishment of close interactions between family members [24].

The use of the collaborative approach and "familiness" leads to more customized knowledge-based solutions that are triggered by trust, reciprocity, and cultural uniformity within family businesses. Higher trust is often associated with a perception of reliability [25] and competence [26] and triggers reciprocation of transfers by encouraging knowledge transmitters to send additional knowledge in return for the knowledge received. Moreover, trust—which is one of the building blocks of the family business culture—leads to the creation of a social community in which knowledge transfer is facilitated [24]. Because of their unique characteristics, family businesses succeed in transferring tacit knowledge which is action-centered and depends on individual know-how and experience. Tacit knowledge is a predominantly social process that is incorporated in business routines [27,28] and is transferred through interaction and social capital readily available in family businesses [29]. These can increase the depth, scope, and efficiency of tacit knowledge transfers through close social interactions [26,30], mutual beliefs, and shared paradigms that promote common understandings and contribute to the formation of high levels of social capital [31]. Contrasting with the transfer of intangible and tacit knowledge, the transfer of explicit knowledge is easier to achieve as it does not rely on social capital and close relationships.

### 2.1.2. Knowledge Assimilation

The effective update of a firm's dynamic capabilities and routines cannot be achieved without knowledge access and protection. The newly acquired knowledge must be retained to lead to value creation [32]. Many terms have been used in the literature to describe knowledge assimilation, such as knowledge embodiment [33], knowledge retention, knowledge codification [34], and organizational memory [35]. The objective of knowledge assimilation is to make knowledge available and accessible to decision makers [36].

Family firms must increase their problem-solving abilities and their capacity to update their strategies within a changing environment. Stored knowledge and existing routines will be used as input for decision making and value-creating activities [37]. Information technology solutions help capture and store codified knowledge and allow easy and quick access to knowledge repositories. In the case of small- and medium-sized family firms, due to the centralization of decision making in the hands of owner-managers, information systems may not be very sophisticated. Family firms tend at the same time to formalize

their knowledge transfer processes, which can make the exchange of tacit knowledge more challenging. The firm's stored knowledge is retained in the organizational memory at the individual and collective levels [38]. At the individual level, memory is linked to cognitive maps and mental models, whereas it is incorporated in cultural values and beliefs at the collective level [37]. Organizational memory can also be observed in the social networks through which knowledge is acquired, combined, and communicated to others to create a new reality.

The uniqueness of the family characteristics in regard to its social capital and network facilitates the retention of knowledge in the organizational memory, yet might pose some challenges in the retrieval and transformation of existing knowledge [24]. The uniformity of cognitive maps and mental models in family businesses impedes the effective retrieval and transformation of stored knowledge. Knowledge loss, which is characterized by personnel withdrawal from the firm which impacts organizational memory and destroys or damages individual repositories, team coordination routines, and social networks, is less present in family businesses because of the stability of their social network and the existence of strong family ties and trust [38,39]. Unlearning by voluntarily dismissing the outdated knowledge that is not anymore aligned with the firm's strategies and which can be considered as an opportunity to update the firm's core values, beliefs, and norms that guide employee behaviors [40] can be challenging for small- and medium-sized family firms, as these structures tend to retain tacit knowledge and suffer from family groupthink as they experience difficulties to renew and update their existent stock of knowledge [39].

### 2.1.3. Knowledge Transformation

Knowledge creation depends on the constant interaction between four conversion modes—tacit to tacit, explicit to explicit, tacit to explicit, and explicit to tacit—which allow knowledge to be detached from individuals and transformed into organizational knowledge [41,42]. The first conversion mode, from tacit to tacit, or socialization, enables employees to transfer tacit knowledge without changing its format. For example, apprentices learn craftsmanship through observation, practice, and imitation. In this context, knowledge transfer depends on skills and shared experiences rather than on the capacity to articulate the tacit knowledge [42]. When knowledge senders and recipients share a common knowledge base and when knowledge recipients possess the skills that enable them to learn through practice, knowledge can be transferred without changing its tacit nature. This conversion process is embedded into the specific context of the apprenticeship and is difficult to replicate in other situations [41]. The second knowledge conversion mode, from explicit to explicit or combination, allows new knowledge to be created through the combination of different pieces of explicit knowledge. The newly created knowledge is explicit, and less context-dependent than the knowledge created through socialization. The third conversion mode, externalization, allows a change in the nature of the knowledge from tacit to explicit [41]. Externalization is triggered by successive rounds of meaningful dialogues between employees that are activated by social interaction mechanisms [42]. The use of metaphors reveals hidden tacit knowledge and leads to the creation of new explicit knowledge. The fourth knowledge conversion mode, internalization, leads to the transformation of explicit knowledge into tacit knowledge. Experimentation allows employees to transform codified knowledge into individual knowledge and to make it their own, and contributes to the formation of new mental models and cognitive maps which are stored in the memory and are founding elements of individual expertise [42].

The high levels of trust and social capital in small- and medium-sized family firms increase the individual motivation to share knowledge [43]. Small- and medium-sized family firms possess the appropriate structure, context, and processes that facilitate knowledge transfer and transformation [44,45]. A smaller structure, a unique set of skills, culture, social capital, and processes facilitate the knowledge transfer and transformation.

### 2.1.4. Knowledge Exploitation

Different terms have been used to refer to knowledge exploitation in the KM literature. These terms include knowledge leverage [13], knowledge use [13,46], and knowledge utilization [47]. Knowledge exploitation leads to the deployment of already absorbed and transformed knowledge [48] and consists of leveraging the knowledge that has been absorbed and retained in the firm's knowledge base [47]. Knowledge exploitation helps refine and extend existent routines and revise them by incorporating knowledge into the firm's operations. Successful knowledge exploitation is an essential part of the process of knowledge value creation. In family businesses, the owner-managers must exploit the firm's existing knowledge in a way that enables the business to strategically take advantage of available opportunities. Owner attitudes towards risk and capabilities of using existent knowledge and taking a long-term orientation toward exploiting the firm's knowledge resources to develop sustainable capabilities determine how knowledge resources are leveraged and exploited to generate a competitive advantage [24]. Family business owners' belief in the possible consequences of growth influences their attitude towards growth and business strategy [25].

### 2.1.5. Potential and Realized Absorptive Capacities

As external knowledge is critical for value creation, absorptive capacity (ACAP) is an important contributor to a firm's ability to create new knowledge [49] and to absorb newly acquired knowledge. Ref. [14] described ACAP as the ability of a firm to identify and acknowledge the value of new external information, assimilate it, and apply it to commercial ends. ACAP depends on a firm's level of prior related knowledge and is essential in building organizational innovative capabilities [14]. Ref. [15] expanded [14]'s definition and defined ACAP as a set of business routines and processes by which organizations acquire, assimilate, transform, and exploit knowledge to produce dynamic capabilities that improve their ability to gain and sustain a competitive advantage [15] (p. 185). Developing and sustaining absorptive capacity is essential to a firm's long-term success as it can reinforce, expand, or reshape the firm's knowledge resources [49].

Because knowledge creation is not the only source of value creation, firms must rely on the acquisition of knowledge from external sources to expand their knowledge base [50,51], especially when they operate in dynamic environments [52]. This points to the importance of absorptive capacity in helping firms acquire external knowledge and use it for strategic purposes [53]. Small- and medium-sized family businesses have a limited ACAP because of their limited size of activities and customer base. Because of the owner's influence on the firm's strategic direction and overall decisions, family firms can strengthen their absorptive capacities by recruiting professionals who contribute new perspectives, skills, and experience. By behaving like non-family firms and professionalizing their activities, small- and medium-sized family businesses can manage knowledge processes more effectively and make a better use of their existing ACAPs.

Knowledge value creation is not only contingent on the individual absorptive capacity of each employee but also on the firm's capacity to retain external knowledge over time [54,55] so that knowledge processes remain active and updated. This capacity is referred to as a firm's potential absorptive capacity [56]. Developing potential capacity requires knowledge sharing through social integration mechanisms and systematic transfers to strengthen mutual understanding [57,58] and collective knowledge. Firms with an extensive internal knowledge base and with enough experience in external knowledge retention have higher potential capacities. Conversely, low social capital makes the acquisition and assimilation of tacit knowledge more difficult and the expansion of the firm's potential ACAP more challenging and less systematic. A firm's realized absorptive capacity lies in its ability to transform and exploit the knowledge acquired and assimilated within the firm [15]. It is this realized capacity that leads to the application of knowledge through the firm's routines and to the creation of new routines that are aligned with business strategy. However, firms focusing too much on knowledge transformation and exploitation achieve

higher knowledge application rates but fall into a competence trap [59,60], as they often lack the ability to assess their knowledge needs and to acquire external knowledge due to their low potential ACAP. These firms lack the capacity to create dynamic capabilities and the ability to renew, augment, and adapt their core competencies over time [61,62]. Conversely, firms that focus extensively on knowledge acquisition and assimilation are able to renew their knowledge stock and to update their knowledge base as they continue to expand their potential ACAP [15,63]. However, these firms may suffer from having too much knowledge without gaining the benefits of exploitation [15].

With their developed social network and shared cognitive maps, small- and medium-sized family firms favor the use of a more conservative approach in dealing with knowledge resources and prefer to focus on the transformation, transfer, and use of existing knowledge rather than to seek new external knowledge and revise existing business strategies [24]. This approach emphasizes the exploitation of the existing knowledge base rather than investing in a new stock of knowledge. Exploratory knowledge that relates to the potential ACAP targets the development of new mental models that are used to revise actual business strategy and align it with market conditions [64], whereas exploitative knowledge that relates to the realized ACAP addresses today's competitive advantage and is built on existing procedures and routines [65]. Often, family firms are not capable of managing the tension between knowledge exploration and expansion of the knowledge base to discover new sources of competitive advantage, or the tension between knowledge exploitation to execute today's strategies and manage the existing competitive advantage [24,25]. Moreover, small- and medium-sized family firms are not able to assess how much efforts and resources they need to invest in knowledge acquisition, fail to properly value what they already know, and question their established knowledge routines only in times of crisis [65]. Because of their conservative approach, limited resources, and cultural uniformity, small- and medium-sized family firms might fail to manage the tension between exploring new business routines and using old certainties, and might be unable to see the threats they face and respond to changes in the environment [66].

## 3. Research Design and Methodology

### 3.1. Multiple Case Study Analysis

Qualitative research gives a deeper understanding of phenomena such as knowledge management processes and absorptive capacities in small- and medium-sized family firms. This is why we used qualitative research here rather than quantitative [67,68]. Through a multiple case study approach, we studied the patterns in different family businesses dealing with the concepts of knowledge management processes and absorptive capacity. Multiple case studies enhance construct validity by building insights and considering contextual factors [69,70].

### 3.2. Case Selection

The firm size of small- and medium enterprises is measured in a variety of ways. While the number of employees, sales, assets, and industrial classification are typically used to determine a firm's size, the various specificities of economies make it difficult to adopt one single definition. We used the definition of Dubai SME to select our case firms. Dubai SME defines small businesses in the manufacturing and service industries as firms employing less than 100 employees, whereas medium businesses have up to 250 employees. We followed Eisenhardt (1989) to decide on the number of case studies and made sure they are consistent with similar published studies in the fields of knowledge management and family business. The case studies conducted aimed to analyze KM processes and ACAP in selected firms, so the level of analysis is the organization. We used theoretical sampling to improve external validity [69] by analyzing cases that represent many theoretically defined factors in the KM and family business literature. As a result, we estimate that these case studies will produce contrasting results but for predictable reasons leading to theoretical reproductions [69] (pp. 46–53). As knowledge resources are context-driven, we selected

companies from different sectors of activity operating as family businesses. Four steps were used for case selection. The first requirement for selection was that the company had a local presence in the UAE. The second criterion was the continuity of operations with a minimum presence of five years in the market. The third criterion was that it operated as a family business. The fourth criterion was that it had less than 200 employees.

Four companies were identified and invited to participate in the study. However, we were unable to gather reliable data from two companies due to difficulties in accessing the managers or business owners. Our research thus relies on two in-depth case studies. This is in line with [71], who recommended two to six cases for theory building. To guarantee respondent anonymity, we used a generic name for each case firm. The two case firms are referred to as the gypsum company (GYPSCO) and the waste management company (WASTCO). Table 1 provides the case study breakdown, along with key information on the two family firms.

**Table 1.** Case study breakdown and company profiles.

| Label | GYPSCO | WASTCO |
|---|---|---|
| Business segments | Construction | Waste Management |
| Size (FTEs) | 43 | 195 |
| Country of origin | UAE | UAE |
| HQ | Sharjah, UAE | Dubai, UAE |
| Number of offices | 1 | 3 |
| Board of directors/Chairman | No | Yes/Father |
| Generation/Management | Third generation/3 brothers | Third generation/4 brothers |
| Year of creation of case company | 1977 | 1996 |
| Managers interviewed | Business owner 1 Business owner 2 General Manager | Business owner 1 Business owner 2 General Manager |
| Part of a group of Family Businesses | No | Yes |
| Other activities of the group | None | Three divisions |
| Year of creation of the company/group | 1977 | 1947 |

### 3.3. Data Collection and Analysis

Data were collected through in-depth semi-structured interviews with the family business owners and managers. The interviews were conducted in January and February 2020. The questionnaire included questions covering the knowledge management processes, the potential and realized absorptive capacities, and the characteristics of the family business. We pretested and validated the questionnaire with two experts in the fields of KM and family business. We developed a case protocol for face-to-face interviews to ensure consistency. We interviewed the general managers and business owners of the two family businesses. In family businesses, the owners are the managers, as they take an active role in determining business strategy and improving the competitiveness of the business. All of them were selected because of their knowledge of business strategies and knowledge resources. Different perspectives were collected through face-to-face interviews with the different respondents in each case firm [72]. Each interview lasted between 60 and 90 min. Data were collected by different interviewers in the two companies. Three respondents from each family firm were interviewed. The interviews were taped and then transcribed. For coding purposes, we used a content analysis framework [73]. We developed category systems for the KM processes and absorptive capacities in a family business context. The adopted category systems for data coding were designed based on existing literature in

KM and family business. For example, the KM literature identified several knowledge transformation strategies such as externalization and internalization, and different types of knowledge (e.g., tacit and explicit) and absorptive capacities [15]. To meet the prerequisites of grounded research, our inductive qualitative research used open-ended questions and a flexible approach to data collection. This led to the exploration of emerging themes and to the possibility of discovering concepts beyond those studied in the KM and family business literature.

We introduced the study to the respondents, along with the research questions, and provided an overview of the concepts covered in the questionnaire. Our research instrument covered these areas of investigation:

- The primary KM processes (acquisition, assimilation, transformation, and exploitation);
- The potential and realized absorptive capacities; and
- The characteristics of the family business context (managerial roles of the family members, role of trust, importance of social capital, etc.).

A selected number of interview questions is presented in Appendix A. The coding of the data identified the extent to which practices related to knowledge resources were implemented in the investigated companies. Each case study was individually coded, and then compared to the results of the coding protocol to ensure consistency. Coding was only considered complete when we reached a consensus on each construct. We started the cross-case analysis by looking at similarities and differences between cases after we analyzed the individual case studies. Our objective was to generalize beyond the data and, through this, discover how some key factors affect the management of knowledge resources in small- and medium-sized family businesses. Finally, we compared our results to the theoretical insights and empirical findings contained in the literature on KM and family business [71]. The next section provides the results of the study.

## 4. Results: Within-Case Analyses

GYPSCO specializes in gypsum works for small and midsize luxury non-commercial real estate projects with a focus on craftsmanship and customization. The firm is managed by three brothers representing the third generation of owner-managers. The business operates through word of mouth and does not use marketing or social media to reach out to customers. The overall ACAP of the business is low since the laborers are uneducated and only office staff have degrees.

WASTCO was founded in 1996. It operates in the business of waste management solutions such as industrial cleaning services and products. Four brothers manage the four divisions that are part of the family business group. A division specialized in real estate, a tire division, a waste management division, and a cultural foundation are part of the group. WASTCO has a board of directors formed by the four brothers and their father who acts as the chairman of the board. The parent group of companies was established by the chairman more than 50 years ago. WASTCO employs 195 employees in the UAE and its target customers are hotels and restaurants. Table 2 provides a summary of the within-case analyses.

**Table 2.** Within-case analyses.

|  | Case 1: GYPSCO | Case 2: WASTCO |
| --- | --- | --- |
| Knowledge Acquisition | <ul><li>Word of mouth and no customer development</li><li>Knowledge to prequalify and bid for projects</li><li>Face to face as much as possible, then email and WhatsApp</li><li>Knowledge from family members working in the same industry</li></ul> | <ul><li>Health and safety regulations</li><li>Owners attend seminars and industry events</li><li>Knowledge about customers and suppliers</li><li>Customer visits to develop their ACAP and increase adoption</li><li>Knowledge for customers</li></ul> |

**Table 2.** *Cont.*

| | Case 1: GYPSCO | Case 2: WASTCO |
|---|---|---|
| Knowledge Assimilation | <ul><li>Minimum reliance on paperwork</li><li>Knowledge retained at individual level</li><li>Owners and a few knowledgeable and experienced employees</li><li>No information systems</li><li>No knowledge codification and no manuals or guidelines</li><li>Tacit knowledge</li><li>Knowledge-intensive business</li></ul> | <ul><li>Work processes are automated</li><li>SOPs and policies</li><li>Databases to store codified knowledge</li><li>Knowledge sharing through weekly departmental meetings</li><li>Cross functional teams</li><li>Regular meetings between HoDs and top managers to monitor strategy implementation progress</li><li>Knowledge assimilation at the individual and collective levels in the different departments</li></ul> |
| Knowledge Transformation | <ul><li>Learning by doing</li><li>Stability of know-how and routines despite technological advancements in the industry</li><li>Knowledge is tacit and is passed on from employee to employee</li><li>Experienced workers for complex tasks</li><li>Unqualified workers for basic tasks</li><li>No training and development as owners fear losing employees to competitors</li></ul> | <ul><li>Knowledge in databases is regularly updated</li><li>A wall of shame is used to advertise failures internally</li><li>Knowledge is transformed at the individual and collective levels</li></ul> |
| Knowledge Exploitation | <ul><li>No growth because of adopted KM approach</li><li>Value lies in specialization and protection of know-how</li><li>Competitive advantage depends on exploitation of existing knowledge and routine stability</li><li>Second generation of owners (the father) has the last say when decisions have to be made or when conflicts need to be solved</li><li>No intervention of second generation in day-to-day activities</li></ul> | <ul><li>Growing market with no major competition</li><li>Knowledge feeds strategy</li><li>Knowledge is used to identify new solutions, improve product mix, and improve service quality</li><li>The business has been restructured a year ago to streamline and align strategy, structure, and processes</li><li>Feed business strategy with newly acquired knowledge</li></ul> |
| Absorptive Capacity | <ul><li>Low overall ACAP despite know-how of owner-managers and senior employees</li><li>Majority of employees is uneducated</li><li>Focus on realized ACAP</li><li>Loyalty and commitment of senior employees help keep the ACAP high</li><li>No employee knows the entire sequence of work to avoid knowledge leakage</li><li>ACAP had decreased overtime as employees want to learn faster but quality of work has decreased because customers value cost reduction over quality</li></ul> | <ul><li>Managers have at least a bachelor's degree, top managers have master's degrees, operatives have technical degrees, and helpers have no degree</li><li>Training and development and knowledge exchange initiatives to increase ACAP</li><li>High retention by building trust, fulfilling personal needs, and offering the highest salaries in the industry</li><li>Career reorientation is used to retain talent</li><li>Equal investment into potential and realized ACAPs</li></ul> |

## 5. Cross-Case Analyses

### 5.1. The Tacit Knowledge Management Approach

The tacit approach emphasizes knowledge assimilation at the individual level and the transformation of tacit knowledge through socialization, externalization, and internalization. Knowledge is passed on from the individual level to the collective level through these different conversion modes [41]. After it is enriched through interaction, knowledge is absorbed and assimilated at the individual level and becomes part of organizational routines [74]. Learning by doing allows the transfer of knowledge through socialization, while conversations allow the externalization of tacit knowledge and the creation of a new reality through the combination of different pieces of explicit and tacit knowledge [75]. In small- and medium-sized family firms, it is through the strong family ties that tacit knowledge and experiential learning are generated. Despite the difficulty of sharing tacit knowledge, relational capital in family firms sets powerful informal knowledge-sharing mechanisms. Frequent face-to-face meetings and high levels of shared beliefs and values lead to effective informal knowledge-sharing practices, whereas "familiness" is used to generate a competitive advantage and to create inimitable resources and capabilities.

GYPSCO protects its know-how from being leaked to the competition by avoiding task delegation and relying on its most experienced employees for the most complicated tasks that require mastery and expertise. Tacit knowledge is passed on to employees in small sequences to avoid having them understand the whole work process. Tacit knowledge is not transferred to junior employees to prevent them from gaining know-how that competitors can benefit from if they decide to leave the family business. As mentioned by Business Owner 1: "We make sure junior employees don't learn a lot so that they take that knowledge and leave the company". Since the family business is known in the industry for its craftsmanship, the younger generation of employees uses the experience they gained in the family business as leverage to negotiate better salaries in competing firms. By protecting the firm's tacit knowledge, the owners choose to keep the firm's scale of activities small. According to the owners, the firm's internal focus and small size have allowed the family business to absorb market shocks much better than the competition. The family business has a stable workforce and did not lay off any employee since its creation in 1977. Small family firms such as GYPSCO are more informally structured and are based on more socially constructed interactions. Knowledge management processes are less established in these structures due to the lack of resources, whereas more attention is given to tacit knowledge and to knowledge sharing in the family firm. At GYPSCO, the technical workforce is divided into helpers, fixers, and finishers. The finishers are the most valuable employees, since their work requires knowledge and expertise and has an artistic component. The retention of such employees is critical for the continuity of business operations. According to Business Owner 2: "The finishers are the most valuable among all employees since they hold the most critical know-how. The family business invests cares about their retention". At GYPSCO, the transfer of tacit knowledge among senior employees is facilitated by the loyalty and trust that are shared between these employees and the owners. Such relational flow enhances the transmission of tacit knowledge and promotes tacit-to-tacit knowledge transfer.

In order for the tacit KM approach to be successfully implemented, a firm's ACAP has to be high within an enabling organizational context. However, by focusing too much on its existing tacit knowledge resources, a family firm might lack the capability to estimate the amount of new knowledge it needs to acquire to augment its knowledge base and potential ACAP. In this context, a firm business might overestimate the importance of its existing routines and accumulated tacit knowledge for competitive advantage and underestimate the role of external forces in shaping business strategies and routines. GYPSCO is an example of such a firm. GYPSCO is unable to manage the tension between potential and realized ACAP, and between the efforts invested in the acquisition of new and market-driven knowledge, and those invested in the exploitation of existing knowledge and reinforcement of realized ACAP and business routines. The difficulty for GYPSCO lies in

the firm's incapacity to adopt a proactive approach to understand market shifts, and to update its competitive advantage and invest in strategy updates instead of focusing on today's successes. The firm's capacity to update its routines and create dynamic capabilities is impaired by its "inward" rather than "market"-driven KM vision. Business intelligence and benchmarking are less emphasized in the tacit KM approach than internal knowledge dissemination and transfer. Family firms might lose ground to competitors if they do not properly manage the tension between potential and realized ACAP and focus on increasing their knowledge base through more knowledge acquisition efforts and the revision of existing routines to generate dynamic capabilities and value. At GYPSCO, the tacit approach has also led to core rigidities and to GYPSCO's incapacity to react to market forces such as changing customer needs and technologies. In fact, despite the fact that GYPSCO's owners attend trade exhibitions to learn about new work methods and technologies, the family firm failed to translate this collected intelligence into business strategy and new routines because of the conservatism of its owners and their overemphasis on stability and existing routines and know-how. Small family firms such as GYPSCO would benefit from a recombination and development of their current knowledge base through the adoption of different KM approaches.

Because of its scale of activity and limited resources, the adoption of new innovations is limited at GYPSCO. Due to the conservatism of its owner-managers, GYPSCO is more risk-averse and less growth-oriented, making any future growth difficult. Withdrawal from reality leads to strategic conservatism in family businesses. Core rigidities emerged at GYPSCO and uncovered its inability to question business strategy and revise existing routines. While still successful in its niche market, GYPSCO focuses on its past and present rather than on the future. Small family businesses like GYPSCO lack the critical size, resources, structure, and vision that would allow them to grow. Trust, stability, context-dependency, and shared mental models become sources of core rigidities and organizational inertia [76]. Although knowledge sharing is critical, family firms have different characteristics potentially hindering knowledge transfer [24]. More specifically, a family firm's most important knowledge lies often in a few closely related family members [25]. This was not the case of GYPSCO, which extended the family ties to a few long-term employees who benefited from the owner's trust and were included in the family social network. The concentration of tacit knowledge and know-how in the family business increases the consolidation of power and control in the hands of those who share the same mental models and family values. Overall, the limitation imposed on knowledge processes led to limited knowledge transfer initiatives, to a focus on internal knowledge processes, and to avoiding externally driven change that could be a source of growth opportunities.

### 5.2. The Strategic Knowledge Management Approach

The strategic KM approach calls for the continuous update and the alignment of business strategy with the changing forces in the firm's external environment. A successful adoption of the strategic KM approach also requires the alignment of internal processes and resources with business strategy. Business intelligence and benchmarking allow family firms to design and implement preventive strategies to protect competitive advantage and to understand the shifts in the market [77]. Market conditions determine the extent to which firms should balance investing in knowledge codification through IT infrastructure and tools, and investing in knowledge management strategies that favor tacit knowledge exchange through mentorship and apprenticeship initiatives. With the adoption of the strategic KM approach, firms acknowledge the effect of globalization on business strategy and perceive strategy development as an externally rather than internally driven process [8]. In this approach, the firm's ACAP is calibrated based on market forces which determine the extent to which investments in knowledge resources should be directed at improving the firm's potential ACAP or its realized ACAP. The renewal or revision of work routines depends on the degree of alignment of internal forces with the market conditions, and on the degree of flexibility of such routines as environmental forces keep evolving.

The case firm WASTCO adopted a market approach to manage its knowledge resources. Despite its high degree of automation, the family business sets strategies to codify knowledge and to encourage knowledge transfers among employees and departments. The cross-functional and weekly departmental meetings, along with the newly launched "think new" initiative, encourage knowledge transformation at the individual and collective levels, as well as tacit knowledge exchanges. The firm's General Manager emphasized: "Although many business operations are codified, we take initiatives to encourage knowledge sharing at the department level and between departments". Customer development initiatives increase the visibility of WASTCO's products and services in the market and increase brand awareness of existing and potential customers. Although WASTCO is one of the first movers in the recycling industry in the country, it still invests in educating customers through face-to-face, email, and social media interactions. Building trust with customers and understanding their needs is essential to sustain WASTCO's competitive advantage. WASTCO's strategies are influenced by market forces and are revised according to the changes in customer needs, technology, industry regulations, and competitive threats. Despite a centralization of decision making at the level of the owner-managers, the family firm decentralizes its process of knowledge creation by investing in a strong family-oriented culture that promotes knowledge sharing. The case firm focuses on growing its ACAP by improving the quality and intensity of knowledge acquisition efforts. WASTCO improves its dynamic capabilities as it aligns its processes and structure with its environmentally driven business strategies. At WASTCO, the owner-managers understand that their survival in the market depends on their capacity to build an extensive customer base, to innovate by updating its services and product lines, and to focus on growth and profitability. As emphasized by Business Owner 1: "The family business operates in a dynamic and highly competitive environment. We keep our knowledge about customers, products, and services current." Being part of a larger group of family businesses allowed WASTCO's strategies to be more impersonal and detached from the personality of the business owners [78]. Despite the fact that only owner-managers are members of the board of directors, the presence of governance mechanisms at WASTCO supported the adoption of the strategic KM approach.

Family businesses generally dedicate fewer resources to knowledge acquisition efforts as they are less growth-oriented and more conservative in approaching strategic decisions. The market orientation adopted by WASTCO's owner-managers enabled the family business to value knowledge acquisition and to invest in knowledge transfer initiatives. On a similar note, knowledge is often centralized in family firms, and acquisition efforts are heavily influenced by the presence of family members in positions of power who influence the family firm's knowledge management strategies and orientations and are often the victims of their own strategic conservatism. While the firm's strategic direction was heavily influenced by the owner-managers vision at WASTCO, their conservatism was neutralized by the strategic orientation adopted in managing the firm's knowledge resources. The usual overreliance on personal judgement rather than on conducting an objective assessment of the knowledge needs in family businesses was offset in WASTCO by its focus on sustaining its actual competitive advantage and investing in identifying sources of future competitive advantage. This led to a better balance between the family firm's investments in its potential and its realized absorptive capacities.

## 6. Conclusions and Implications

The objectives of our study are threefold: (1) to explore the specificities of the relationship between KM processes and a firm's absorptive capacity in the context of a knowledge-based view using externally and internally generated knowledge as input, which requires further investigation of the role of knowledge processes in value creation and exploring the relationships between the processes and capacities mentioned above; (2) to delineate and characterize the importance of the potential and the realized absorptive capacities; and (3) to examine and explore the knowledge specificities of small- and

medium-sized family businesses based in the United Arab Emirates (UAE). Theoretical contributions are made to the family business literature as this study is grounded in the knowledge-based view approach that enhances our understanding of how small- and medium-sized family businesses manage knowledge processes and absorptive capacities to generate dynamic capabilities. Specifically, this approach captures the contribution of knowledge resources to competitive advantage and sheds light on how "familiness" acts as an advantage or disadvantage in the context of the management of knowledge resources in small- and medium-sized family businesses. Since the resource-based view does not fully capture the contribution of the behavioral and social aspects of family businesses to the generation of dynamic capabilities, the knowledge-based view adds value by enhancing our understanding of how knowledge processes and absorptive capacities are impacted by the characteristics of small- and medium-sized family businesses. From a theoretical standpoint, the study provides a better understanding of how owners' personalities and their strategic orientation, as well as the concentration of knowledge in the hands of a few family members and employees, influence the knowledge management processes adopted in family businesses.

Despite the fact that family businesses are often described as having unique structural advantages over non-family businesses, as they benefit from high social capital and the presence of fluid socio-cultural knowledge practices that enhance internal knowledge transfers and contribute to improving the family firm's capacity to manage knowledge resources, they often fail to balance their potential and realized ACAPs and to manage the efforts invested in knowledge exploration and exploitation. Family businesses that invest more resources in expanding their knowledge base focus on future competitive advantage rather than on existing knowledge routines, whereas those that invest more resources to increase knowledge exploitation focus on today's routines rather than on future customer value creation. The dominance of a few family members and their control over core strategic decisions hampers vital knowledge integration mechanisms in small- and medium-sized family firms and negatively impacts their knowledge management processes. The results of this study show that the owner-managers can consciously adopt a more market-oriented knowledge management approach when they value both knowledge exploration and knowledge exploitation and invest in sustaining the firm's actual competitive advantage and in identifying future sources of competitive advantage.

While a family firm's strong relational ties provide powerful informal knowledge-sharing mechanisms that help mobilize tacit knowledge resources that are otherwise difficult to share, knowledge-sharing processes benefit from a higher degree of formalization of knowledge management strategies, which allows family businesses to avoid an over-reliance on tacit knowledge. Business strategies as well as knowledge resources are, in that context, driven by market forces, by the use of the family business structural characteristics to mobilize tacit knowledge, as well as by investments in knowledge acquisition, assimilation, and transformation. The results show that when family businesses lack resources, structure, or technology to compete effectively in the marketplace, they tend to adopt reactive strategies rather than strategies that anticipate market opportunities. This encourages the emergence of family-induced groupthink and exacerbates the concentration of decision-making powers in the hands of owners and a few family members [24]. Survival and preservation of existing competitive advantage become dependent on the family firm's capacity to promote cultural uniformity to align the family firm's strategic orientations with those of owners. Instead of capitalizing on their structural specificities, family businesses often tend to design and implement business strategies that preserve today's competitive advantage rather than address tomorrow's challenges [29]. The results show that family firms that are capable of balancing between knowledge exploration and knowledge exploitation are market-driven and acknowledge that securing a sustainable competitive advantage depends on the family firm's ability to sustain its existing operations and routines, and to invest in improving profitability and growth.

Small- and medium-sized family businesses should target the creation of dynamic rather than rigid capabilities by investing in activities that target knowledge exploration and knowledge exploitation at the same time. Moreover, the success of the knowledge management approach depends on the alignment of the knowledge infrastructure with the knowledge management processes, along with the adoption of an approach that effectively manages the tension between the family firm's potential and realized ACAPs. Knowledge management processes must be paired with the appropriate ACAP to lead to optimal value creation.

## 7. Limitations and Future Avenues of Research

The research limitations could present interesting avenues of research to explore in the future. First, since this study used a multiple case study methodology to investigate how two family firms operating in the UAE manage their knowledge resources, the findings may not be generalized to companies in other contexts and settings. Second, the focus of this research was limited to family businesses with limited governance structures. Future research should focus on studying family businesses with boards of directors composed of family and non-family members. Third, we relied in this research on in-depth semi-structured interviews with the business owners and managers of the case study companies, leading eventually to a potential bias of findings based on interviewees' perceptions. Moreover, more insights on how knowledge resources are processed and managed in family businesses could be gained from longitudinal studies and survey-based research. In a similar vein, further study of the impacts of the knowledge management processes on performance and dynamic capabilities would be needed. Additionally, future research should study the mechanisms that allow firms to move from potential to realized absorptive capacity and the extent to which these capacities contribute to the generation of dynamic capabilities in family businesses. Further studies of the impacts of family business characteristics on the firm's success in managing its knowledge processes and ACAPs would also be recommended. The study of the impact of the family firm's size on the management of knowledge resources will improve our understanding of the impact of "familiness" characteristics on the knowledge management approach and the influence that owners and family members have in small- and medium-sized versus large family businesses. From the resource-based view, family firms were described as structures that possess and manage unique knowledge resources that contribute significantly to their knowledge base and are used to generate and sustain competitive advantage. Finally, future studies should focus on studying the specificities of Arab family businesses in comparison with non-Arab ones to identify the impact of cultural differences on the management of their knowledge resources.

**Funding:** This research received no external funding.

**Informed Consent Statement:** Informed consent was obtained from all subjects involved in the study.

**Conflicts of Interest:** The author declares no conflict of interest.

## Appendix A. Selected Interview Questions

1. Describe the types of knowledge that the business needs the most for strategic and operational purposes.
2. How often does the business update its knowledge? Describe how this is done.
3. To what extent does the company rely on knowledge codification versus knowledge that is communicated from person to person?
4. How are business routines and operations sustained and updated?
5. To what extent are employees and managers educated and hold knowledge and expertise (know-how)?

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
