# Peer review of "Managing Knowledge Resources in Family Firms: Opportunity or Challenge?"

_sustainability, doi:10.3390/su14095087_

Round 1

Reviewer 1 Report

This work is to explore the specificities of the relationship between the knowledge management processes and the potential and realized absorptive capacities in a knowledge- based view context. It advances our understanding of the contributions of the knowledge management process and the potential and realized absorptive capacities in small and medium-sized family firms. However, the following issues need to be thought through and modified as necessary.

  1. Quantitative analysis mathematical model should be reflected in the paper.
  2. The most basic research results should be presented in graphs and tables.
  3. Need to cite and refer to the latest research.

Author Response

This work is to explore the specificities of the relationship between the knowledge management processes and the potential and realized absorptive capacities in a knowledge- based view context. It advances our understanding of the contributions of the knowledge management process and the potential and realized absorptive capacities in small and medium-sized family firms. However, the following issues need to be thought through and modified as necessary.

  1. Quantitative analysis mathematical model should be reflected in the paper.

Response to reviewer: Thank you for your comment. The article is based on a multiple case-study methodology (qualitative research). A quantitative approach could be used in future research to increase robustness.

  1. The most basic research results should be presented in graphs and tables.

Response to reviewer: Thank you for your comment. Table 2 provides a summary of within case analysis. The cross-case analysis is presented in a text format to allow for discussion of findings and reflect on how it complements or opposes previous research

  1. Need to cite and refer to the latest research.

Response to reviewer: Thank you for your comment. The literature review details the most advanced theoretical and empirical findings pertaining to the concepts object of the study. The review is focused and addresses only relevant research. More recent references have been used as reflected in your comment. References have been updated to reflect latest advancements in the family business and knowledge management literatures.

Reviewer 2 Report

Dear authors, after reading your work I have the following recommendations that maybe could help you to improve this work:

The topic of the paper is relevant and current to understand the knowledge management process and the potential and realized absorptive capacities in small and medium-sized family firms.

The paper is well written, the subject address is pertinent, and more research should be done on this topic in the literature. The paper contains new and significant information adequate to justify publication. As such, the paper could have a place in Sustainability editorial line.

The title is specific and relevant.

The abstract has all the fundamental parts for understanding the study, from its framework, methodology and main conclusions.

The keywords are specific to the article.

Is clear the reading of the text (textual coherence and cohesion). 

The introduction is clear about the importance of studying, framing and identifying a research question.

Literature review should be more coherent and flow. The authors must include a section with other empirical studies carried out on this topic and include recent publications in this topic.

The point is to do critical review of literature. Also there should be more relevant articles included published in the last several years.

In this article, family involvement in ownership and management is discussed, but “family involvement in governance” is not addressed and this could be of great relevance to complement this article.

In the methodology the authors must reveal when the interviews were taken (place the study in time).  

In the literature, although qualitative research is quite important, it is true that several studies recommended that this approach should be complemented with a quantitative analysis. In your case, could your primary information (obtained from the interviews) to be complemented with secondary information to provide more robustness?

The research results must be improved. The interviews with people took 90 minutes, but their thoughts and ideas are not presented anywhere. You should examine a highly impactful article and follow a similar style of delivery. The results should be more structured, and possibly include quotes of the management.

The conclusions adequately tie together the other elements of the paper.

I suggest the authors include an appendix with the interview’s questions.

I hope that my comments and suggestions can help to improve the paper.

Thank you for the opportunity to read your article.

I wish all the best to the authors!

Author Response

Dear authors, after reading your work I have the following recommendations that maybe could help you to improve this work:

  • The topic of the paper is relevant and current to understand the knowledge management process and the potential and realized absorptive capacities in small and medium-sized family firms.
  • The paper is well written, the subject address is pertinent, and more research should be done on this topic in the literature. The paper contains new and significant information adequate to justify publication. As such, the paper could have a place in Sustainability editorial line.
  • The title is specific and relevant.
  • The abstract has all the fundamental parts for understanding the study, from its framework, methodology and main conclusions.
  • The keywords are specific to the article.
  • Is clear the reading of the text (textual coherence and cohesion). 
  • The introduction is clear about the importance of studying, framing and identifying a research question.

Response to reviewer: Thank you for the above comments

  • Literature review should be more coherent and flow. The authors must include a section with other empirical studies carried out on this topic and include recent publications in this topic. The point is to do critical review of literature. Also, there should be more relevant articles included published in the last several years.

Response to reviewer: Thank you for your comment. The literature review details the most advanced theoretical and empirical findings pertaining to the concepts object of the study. The review is focused and addresses only relevant research. More recent references have been used as reflected in your comment.

  • In this article, family involvement in ownership and management is discussed, but “family involvement in governance” is not addressed and this could be of great relevance to complement this article.

Response to reviewer: Thank you for your comment. The paper is not about family involvement in governance since the paper studies how knowledge resources are managed in family firms rather than family involvement in governance. The latter could be explored in another research on family firms.

  • In the methodology, the authors must reveal when the interviews were taken (place the study in time).  

Response to reviewer: Thank you for your comment. The timeline of the interviews is now mentioned in the methodology section.

  • In the literature, although qualitative research is quite important, it is true that several studies recommended that this approach should be complemented with a quantitative analysis. In your case, could your primary information (obtained from the interviews) to be complemented with secondary information to provide more robustness?

Response to reviewer: Thank you for your comment. The article is based on a multiple case-study methodology (qualitative research). A quantitative approach could be used in future research to increase robustness.

  • The research results must be improved. The interviews with people took 90 minutes, but their thoughts and ideas are not presented anywhere. You should examine a highly impactful article and follow a similar style of delivery. The results should be more structured, and possibly include quotes of the management.

Response to reviewer: Thank you for your comments. The results section has been improved and quotes from interviewees included to enrich the findings.

  • The conclusions adequately tie together the other elements of the paper.

Response to reviewer: Thank you for your comment.

  • I suggest the authors include an appendix with the interview’s questions.

Response to reviewer: Thank you for your comment. Sample interview questions are presented in an appendix as suggested.

I hope that my comments and suggestions can help to improve the paper. Thank you for the opportunity to read your article. I wish all the best to the authors!

Response to reviewer: Thank you for your comments.

Reviewer 3 Report

I am thanking the author for his contribution, development and clear presentation of the paper. I think it is right to point out that small and medium-sized family businesses could improve their absorption capacity with the external expertise of professionals. I Know that the multiple case studies increase validity through the development of insights and consideration of context-dependency, but the paper could include some statistical method that would increase the validity of the results of the qualitative study. The criteria for the selection of cases and the treatment of cross-referenced information could be adequate. The research limitations could present opportunities and avenues to explore in future research. It is a preliminary research that is close to reality but gives many opportunities to continue in this line of research. 

Author Response

I am thanking the author for his contribution, development and clear presentation of the paper. I think it is right to point out that small and medium-sized family businesses could improve their absorption capacity with the external expertise of professionals. I know that the multiple case studies increase validity through the development of insights and consideration of context-dependency, but the paper could include some statistical method that would increase the validity of the results of the qualitative study. The criteria for the selection of cases and the treatment of cross-referenced information could be adequate. The research limitations could present opportunities and avenues to explore in future research. It is a preliminary research that is close to reality but gives many opportunities to continue in this line of research. 

Response to reviewer: Thank you for your comment. The article is based on a multiple case-study methodology (qualitative research) and uses extensive semi-structured interviewees to collect insight from family business owners and managers. A quantitative approach could be used in future research to increase robustness.

Round 2

Reviewer 1 Report

The authors have responded to my questions with some modifications.